# Effects of Androgen Treatment on Growth in Patients with 5-α-Reductase Type 2 Deficiency

**DOI:** 10.3390/jpm13060992

**Published:** 2023-06-13

**Authors:** Hae In Lee, Sujin Kim, Sang-woon Kim, Myeongseob Lee, Kyungchul Song, Junghwan Suh, Yong Seung Lee, Hyun Wook Chae, Ho-Seong Kim, Sangwon Han, Ahreum Kwon

**Affiliations:** 1Department of Pediatrics, CHA Gangnam Medical Center, CHA University, Seoul 03722, Republic of Korea; 08orange13@gmail.com; 2Department of Pediatrics, Severance Children’s Hospital, Institute of Endocrinology, Yonsei University College of Medicine, Seoul 03722, Republic of Korea; nounour89@yuhs.ac (S.K.); mslee8791@yuhs.ac (M.L.); endosong@yuhs.ac (K.S.); suh30507@yuhs.ac (J.S.); hopechae@yuhs.ac (H.W.C.); kimho@yuhs.ac (H.-S.K.); 3Department of Urology, Urological Science Institute, Yonsei University College of Medicine, Seoul 03722, Republic of Korea; khisin@yuhs.ac (S.-w.K.); asforthelord@yuhs.ac (Y.S.L.); 4Department of Urology, CHA Gangnam Medical Center, CHA University, Seoul 03722, Republic of Korea; swhan123@chamc.co.kr

**Keywords:** 5-α-reductase type 2 deficiency, dihydrotestosterone, testosterone enanthate, bone age, standard deviation score

## Abstract

Background: Patients with 5-α-reductase type 2 deficiency (5αRD2) require androgen treatment for the growth of normal male external genitalia. Since limited research has been conducted on the effects of androgen treatment on height in individuals with 5αRD2, we investigated the effect of androgen treatment on bone age (BA) and the height status in children with 5αRD2. Methods: Of the 19 participants who were followed up for an average of 10.6 years, 12 received androgen treatment. BA and height standard deviation scores (SDS) were compared between the treatment and non-treatment groups, as well as between the dihydrotestosterone (DHT) and testosterone enanthate (TE) treatment groups. Results: Despite the above-average height of the 19 patients with 5αRD2, the height SDS relative to BA (htSDS-BA) was below average, particularly in the androgen treatment group. DHT treatment did not lead to an increase in BA or htSDS-BA, whereas TE treatment resulted in BA advancement and decreased htSDS-BA, especially in the prepubertal period. Conclusions: DHT treatment is more favorable for height than TE treatment in patients with 5αRD2, particularly during the prepubertal period. Therefore, age and the type of androgen used should be carefully considered to minimize the risk of height reduction in these patient groups.

## 1. Introduction

5-α-Reductase type 2 deficiency (5αRD2, OMIM #607306) is a rare autosomal recessive condition affecting sex development in 46,XY individuals, caused by mutations in the gene *SRD5A2*. This gene encodes for the 5-α-reductase type 2 isozyme, which converts testosterone (T) into dihydrotestosterone (DHT) [1]. DHT is the most potent [2] and principal endogenous androgen in humans and is essential for the normal development of male external genitalia [3]. Impaired conversion of T to DHT is responsible for the symptoms of under-masculinized male external genitalia in patients with 5αRD2. These patients exhibit a wide range of genital ambiguities, from apparent male external genitalia with mild under-masculinization to complete female external genitalia.

Androgen treatment, including testosterone enanthate (TE), is commonly recommended for pediatric patients with under-masculinization, such as those with micropenises. However, the use of TE in children with micropenises is controversial because of concerns regarding sexual precocity and early epiphyseal closure. Although only a small proportion of T (~0.2%) is converted to estradiol, the potency of estradiol is approximately 100-fold higher than that of T [3], which can lead to a potential advancement in bone age (BA) with TE treatment. Consequently, lower doses of TE are recommended for prepubertal patients [4].

In contrast, DHT transdermal gels, which have been available since 1982 (Laboratories Besins International, Paris, France), can be used to increase penile length in patients with 5αRD2 [5]. They have been regarded as the preferred treatment option over TE in patients with 5αRD2 who cannot convert T to DHT. This is owing to their stronger binding affinity to androgen receptors (AR) and their inability to be converted to estrogen by aromatase [2,3].

Despite the potential benefits of androgen treatment in patients with 5αRD2 and micropenises, few studies have evaluated the long-term effects of androgen treatments, including TE and DHT, on the growth of these patients. Understanding the effects of androgen treatment on height in patients with 5αRD2 is crucial for the clinical management of children with this condition. Accordingly, we aimed to assess the longitudinal effects of androgen treatment (TE and DHT transdermal gel) on growth outcomes, including BA, height standard deviation score (SDS) for BA, in 19 participants with 5αRD2. Additionally, we compared the changes in BA and height SDS between patients who did and did not receive androgen treatment and between patients receiving DHT transdermal gel treatment and those receiving TE treatment to investigate the impact of each treatment on growth.

## 2. Materials and Methods

### 2.1. Participants

Nineteen patients with 5αRD2 were enrolled in this study. All participants underwent evaluation at the Disorders of Sex Development Clinic at Severance Children’s Hospital over 20 years (from July 2003 to January 2023). They were diagnosed with 46,XY DSD, and subsequently underwent genetic analysis to identify the underlying mutations. The primary methods used for molecular genetic analysis included: (1) targeted direct sequencing (predominantly used prior to 2015), (2) DSD gene panel-based next-generation sequencing (NGS; used between 2016–2018 and 2020–2023), and (3) whole-exome sequencing (WES), which was performed as part of a study to determine the causative genes of DSD in 2019. Sequencing and bioinformatic analyses were performed as described previously [6]. The annotated mutation records and population frequencies of the called variants were obtained from public databases, including the Human Genome Mutation Database (HGMD) [7], ClinVar [8], the Exome Sequencing Project [9], 1000 Genomes Project [10], Genome Aggregation Database (gnomAD) [11], and the Korean Reference Genome database [12]. Subsequently, we used five in silico prediction algorithms (SIFT, PolyPhen2, FATHMM, and CADD) to predict the deleterious effects of each annotated variant. We selected variants that met the following criteria: (1) an allele frequency of less than 1% in the 1000 Genomes Project and gnomAD; (2) not present in our in-house database; (3) protein-altering variants; and (4) a high read quality (defined as a read number greater than 20 or a quality score [QS] higher than 30). Candidate variants were further confirmed using Sanger sequencing. In cases where the variants were suspected to be compound heterozygotes, we examined the genome of the parents of the participants to determine the inheritance pattern. Finally, the candidate variants were interpreted based on a 5-tier classification system recommended by the American College of Medical Genetics and Genomics and the Association for Molecular Pathology [13]. Two participants were diagnosed with 5αRD2 using direct *SRD5A2* sequencing, and the chromatogram results are presented in Appendix A. NGS was performed on 70 patients with DSD, of whom 10 participants (14.3%) were identified as having 5αRD2. WES was conducted on 74 patients, of whom 7 (9.6%) were diagnosed with 5αRD2. Ultimately, we identified 19 patients with eight different mutations in *SRD5A2* (Appendix A). All patients also underwent a clinical evaluation to confirm their diagnosis. They exhibited ambiguous genitalia or sex incongruity, possessed a 46,XY karyotype, and displayed no abnormalities in the sex-determining region of Y. Furthermore, their adrenal function was normal during the initial evaluation, and the T/DHT ratio was elevated at baseline or after a human chorionic gonadotropin (hCG) stimulation test.

This study was approved by the Institutional Review Board of Severance Hospital Clinical Trial Center (subject no. 4-2022-1646). In a previous study (n = 74) that aimed to identify the genetic cause of 46,XY DSD using WES, informed consent was obtained from the parents of the participating children and the clinic for the collection of genetic information. In addition, the current retrospective study only analyzed the results obtained from the medical records of the clinic; therefore, the requirement for informed consent from other participants was waived. However, owing to the limited number of participants included in the study, patient data collection was conducted on a secure computer, while ensuring the non-disclosure of sensitive personal information by following the recommendations of the Institutional Review Board of Severance Hospital Clinical Trial Center. The current study was conducted in accordance with the Declaration of Helsinki to protect the rights of the participants and personal information.

### 2.2. Anthropometry

Patients had regular follow-up appointments at the outpatient clinic every 3 months to 3 years. The heights and weights of participants were assessed at each visit. Height was measured using a Harpenden Stadiometer (precision, 0.1 cm; the height of patients under 2 years old was measured in a recumbent position). Weight was recorded to the nearest 0.1 kg, using a digital scale with participants dressed in light clothing and without footwear. Growth parameters, including height and weight, were expressed as SDS and calculated using the 2017 Korean Children and Adolescents Growth Chart [14]. Bone age (BA) was assessed according to the Greulich–Pyle method by the same observer [15]. The height and weight SDS for BA were defined as the calculated SDS based on BA and were calculated using the 2017 Korean Children and Adolescents Growth Chart.

### 2.3. Evaluation of Clinical Manifestation

The clinical manifestations of the participants, including their external genital phenotypes, family history, and sex of rearing, were investigated. The external genital phenotype of each participant was evaluated at the first visit using the external genitalia score (EGS) [16], which is the summation of the scores of the external genitalia such as the presence of labioscrotal fusion, genital tubercle length, location of the urethral meatus, and both gonads. A micropenis was defined as a stretched penile length <−2.5 standard deviations (SD) of the same age [17]. Stretched penile length was measured by determining the distance from the pubic symphysis attachment to the tip of the glans while fully stretching the penis. Cryptorchidism was confirmed through physical and ultrasound examinations and classified as either intra-abdominal or inguinal based on the location of the testis. Moreover, hypospadias was subcategorized into perineal, scrotal, penoscrotal, proximal, midshaft, and distal, according to the urethral opening position. Endocrine investigations included measurements of basal luteinizing hormone (LH), basal follicle-stimulating hormone (FSH), basal T, and DHT before and after the hCG stimulation test. Serum LH and FSH concentrations were measured via Chemiluminescent Immunoassay (Atellica^®^ IM 1600, Siemens, Chicago, IL, USA). Serum T concentrations were measured using an electrochemiluminescence immunoassay (Cobas 8000 instrument, Roche Diagnostics, Mannheim, Germany). Additionally, serum DHT concentrations were measured using liquid chromatography-tandem mass spectrometry (Quest Diagnostics, Madison, NJ, USA). The hCG stimulation test was conducted as follows: baseline T and DHT levels were measured at 9:00 a.m. on day 0, after which IVF-C^TM^ (hCG, Seoul, LG Chem Ltd., Seoul, Republic of Korea) was administered at a dose of 1000 IU for three consecutive days (days 0, 1, and 2). On day 3, T and DHT levels were measured at 9:00 a.m. All participants underwent the same assessment.

All except three of the male-assigned participants (N = 12) underwent androgen replacement therapy with DHT cream, testosterone, or both to promote virilization of external genitalia. The dosage and duration of the medication are documented in Appendix A We used 2.5% DHT gel (Andractim^®^, Besins Healthcare, Principauté de Monaco), and one dose of DHT treatment was defined as a dosage of 0.1–0.3 mg/kg/day with a maximum dose of 5 mg/day [18]. Parents were instructed to smear the DHT transdermal gel around the root of the patient’s penis. Before diagnosis with 5αRD2 or when DHT transdermal gel was not available, testosterone was administered. Testosterone was administered using a single dose of TE, Jenasteron^®^, ranging from 25 to 250 mg, either as a single dose or 2–4 consecutive doses at intervals of 2–4 weeks. Hormone administration intervals > 1 year were considered the subsequent cycle, and patients received treatment for 1–4 cycles. Patient data were analyzed by dividing them into non-treatment and post-treatment data. The data from patients who did not receive treatment, as well as the data from patients who received treatment prior to the initiation of the discussed treatment, were classified as non-treatment data. If a new cycle began 2 years before the end of the treatment cycle, the data between the cycles were classified as post-treatment data. However, data that were >2 years after the completion of a cycle were classified as post-treatment data and pre-treatment data for the subsequent cycle until it began.

### 2.4. Statistical Analyses

All statistical analyses were performed using SPSS (Version 26.0; IBM Corp., Armonk, NY, USA). Nonparametric data were presented as median values and ranges (minimum and maximum). Continuous variables were tested for the fitness to normal distribution before a paired t-test was used to compare the difference between non- and post-treatment. Normally distributed data were expressed as mean ± standard deviation (SD). Statistical significance level was set at *p* < 0.05.

## 3. Results

### 3.1. Clinical Characteristics and Status of Androgen Treatment in 19 Patients with 5-α-Reductase Deficiency Type 2

The summary of clinical characteristics of 19 patients with 5αRD2 are presented in Table 1. The median age of the participants during the study was 11.7 years (range 1.9–27.0 years), and they were followed up for a median of 10.6 years (range 1.4–20.0 years) since their initial visit at a median age of 0.6 years (range 7 d–16.4 years). Seven participants (36.8%, Patient nos. 1–7) did not receive androgen replacement therapy. Among them, six participants were initially identified as female. Their median EGS was 2.0 (range 1.0–2.5). Four of the participants (Patient nos. 1–4) underwent bilateral gonadectomy at 4.7 years, 11.1 years, 2.6 years, and 0.6 years old, respectively, and did not receive androgen treatment. Patient 3 underwent feminizing surgery of the external genitalia at 4 years of age; however, the other patients did not undergo such procedures because of their diagnosis and the possibility of gender transition. The other two participants (Patients 5 and 6) requested a gender identity change from female to male after becoming adults. Patient 6 received DHT cream treatments for 4 months; however, this period occurred after the epiphyseal plate had closed. Therefore, the DHT cream treatment did not affect his height. Both patients underwent gender-affirming surgery. Of the 19 patients, Patient 7 presented with a mild isolated micropenis and did not undergo androgen replacement therapy.

The remaining 12 were treated with DHT alone (5, 31.3%), TE alone (3, 15.8%), or DHT and TE (4, 21.1%). The age of treatment, dosage, and duration of the 12 participants are comprehensively described in Appendix A. Twelve patients received a total of 28 cycles of androgen treatment, and among these, there was a good effect on penis size in 14 cycles (50%), while moderate effects were observed in 7 cycles (25%). All treated subjects were followed up and their growth statuses were evaluated for a median duration of 6.6 years (DHT, 5.8 yr, range 1.5–11.0 yr; TE, 5.0 yr, range 1.9–7.9 yr) from the initiation of treatment.

### 3.2. Height SDS and BA in 19 Patients with 5-α-Reductase Deficiency Type 2

The heights measured during each visit were converted into SDS values for the height of BA. The mean height SDS of the 19 participants was ≥0 in non-treatment and after-treatment, and the mean height SDS did not significantly differ between the two groups. However, the difference between BA and chronological age (CA) was >0 in both the non-treatment and post-treatment groups, indicating that BA was more advanced than CA in both groups. Additionally, the difference was significantly higher in the post-treatment group than in the non-treatment group (*p* = 0.022), indicating that BA was more advanced in the post-treatment group than in the non-treatment group. Moreover, the mean height SDS for BA was <0, and the mean height SDS for BA was significantly more decreased in the post-treatment group than that in the non-treatment group (*p* = 0.012, Figure 1). These findings suggest that although the height of patients with 5αRD2 was above average, their height SDS relative to BA was below average owing to advanced BA compared with that of CA. This trend was more pronounced in the post-treatment group than in the non-treatment group.

### 3.3. Changes in the Difference between BA and CA following Androgen Treatment According to Androgen Type

Figure 2A,B presents the changes in BA–CA levels before and after androgen treatment. During DHT treatment, there was minimal change in BA–CA in patients 8, 9, 12, 14, and 15. In patient 13 and the third DHT treatment of patient 14, BA–CA even exhibited a decrease (Figure 2A). Conversely, patients 17, 18, and 19 showed a tendency of stable BA–CA during TE treatment, while patients 13 and 16 experienced an increase in BA–CA during TE treatment (Figure 2B). In other words, DHT treatment did not result in an increase in BA, whereas TE treatment was associated with a tendency for increased BA in some cases.

### 3.4. Changes in the Height SDS for BA following Androgen Treatment According to the Type of Androgen

Figure 2C,D illustrates the changes in height SDS relative to BA following androgen treatment. Except for patient 8 and the fourth DHT treatment of patient 14, all participants who received DHT treatment demonstrated an increase in the height SDS for BA. Patient 8 initially experienced an increase followed by a decrease in height SDS for BA during first treatment, while the second treatment of patient 8 and the fourth treatment of patient 14 showed no change in height SDS for BA (Figure 2C). On the other hand, in the TE treatment group, patients 13 and 16 exhibited a decrease in height SDS for BA. Patient 17 demonstrated minimal change, while patients 18 and 19 experienced an increase in height SDS for BA (Figure 2D). These findings indicate that during DHT treatment, the progressions of BA and CA were similar, leading to a favorable height SDS in relation to BA. Conversely, BA advanced more rapidly than CA during TE treatment, whereas height alterations were not as substantial, resulting in reduced height SDS compared with BA.

### 3.5. Changes in Height SDS for BA Based on the Start of Treatment According to the Age at Treatment

We calculated the change in height SDS for BA based on the time of treatment initiation in relation to age. Among the four patients (patients 9, 12, 13, and 15) who received DHT treatment before the age of 6–7, there was a noticeable increase in height SDS for BA corresponding to the time of treatment initiation. In contrast, patient 8 underwent DHT treatment at the ages of 5.6 and 11.6. At 5.6 years old, there was an increase in height SDS for BA, while at 11.6 years old, the change was minimal. Similarly, patient 14 exhibited an increase in height SDS for BA when treated at 6.1 years old, but showed little change when treated at 7.6 years old (Figure 3A). Conversely, patients 13 and 16 experienced a decrease in height SDS for BA during TE treatment, receiving TE treatment at the ages of 6.5, 9.4, and 6.8 years old, respectively. However, patients 18 and 19, who received TE treatment at the ages of 13.2 and 10.0 years old, respectively, showed an increase in height SDS for BA (Figure 3B). These results suggest that DHT treatment may have a favorable effect on BA and height before the age of 6–7, whereas TE treatment may have a negative effect on BA and height before the age of 10. However, after 10 years of age, TE treatment may have little effect on BA and height, or even have a positive effect.

### 3.6. Comparison of Final Adult Height with Mid-Parental Height in 5-α-Reductase Deficiency Type 2

Of the 19 participants, 6 reached their final adult heights. Among them, four (patients 3, 5, 6, and 7) did not receive androgen treatment, and two received only TE treatment (patients 17 and 18). The final adult heights of the four participants who did not receive androgen treatment were either similar to or greater than MPH, while among the two participants who received TE treatment, one (patient 17) was smaller. The other (patient 18) was taller than the MPH (Figure 4). Patient 17 underwent TE treatment at the age of 10.3 and 11.5. Before starting TE treatment, his BA was already approximately 1.8 years ahead of CA. Although his final adult height was smaller than the MPH, it increased compared with that of the height SDS for BA at the time of treatment initiation (−1.72 at 10.3 years old, −0.59 at 17.9 years old). However, patient 18 underwent TE treatment at the age of 13.2 and 15.8, and the final adult height increased and exceeded the MPH as the height SDS for BA increased after treatment.

## 4. Discussion

A comprehensive understanding of the impact of androgen treatment on the height of individuals diagnosed with 5αRD2 is imperative for its effective clinical management in children. Accordingly, we evaluated the effects of treatment with TE and DHT transdermal gel on the growth outcomes of patients diagnosed with 5αRD2 and observed that the height of 19 patients with 5αRD2 was above average; however, the height SDS relative to BA was below average, particularly in the post-treatment group. This can be attributed to the advancement of BA after androgen treatment. Notably, DHT treatment did not lead to BA advancement, whereas TE treatment exhibited a tendency towards BA advancement. During the DHT treatment, height SDS for BA increased or remained similar, whereas TE treatment resulted in a decrease in height SDS relative to BA. The increase in height SDS for BA during DHT treatment was higher before the age of 6–7 and remained similar at the time of treatment initiation thereafter. Conversely, height SDS for BA during TE treatment was reduced before the age of 10; however, it remained similar or demonstrated a more favorable effect after the age of 10 compared with that during the time of treatment initiation.

The assignment of gender to patients with 5αRD2, which is one of the 46,XY DSD conditions, presents a considerable challenge owing to various factors that need to be considered. Patients with 5αRD2 typically present with vulvas and varying degrees of hypospadias, clitoromegaly, or microphalli; phenotypes that are either mostly male or female are rare [19]. Hence, assigning gender to these patients is crucial. Previous studies have reported that <20% of patients with 5αRD2 were raised as males [20]; however, recent studies have indicated a steady increase in the percentage of males being raised as males, with the number now ranging from 61% [21] to 79% [22]. Furthermore, gender changes from female to male have frequently been reported [23,24]. Virilization that occurs during puberty and exposure to T in the fetal, neonatal, and pubertal stages is believed to play a crucial role in the preference for male gender identity among patients with 5αRD2 [25]. Patients who are under-virilized can benefit from the availability of systematic T or DHT gels, and those with severe micropenis or aphalia can benefit from new genital surgery techniques [25]. The confirmative diagnosis of 5αRD2 has been made easier by improvements in molecular genetic analysis techniques, which has also contributed to an increase in the number of male patients. In the present study, two patients with 5αRD2 changed their gender to male when they reached adulthood.

Given that most patients with 5αRD2 present with clitoromegaly or microphalli, assigning them male gender would require androgen treatment to enlarge the phallus. A review analyzing long-term psychosexual and quality-of-life outcomes in adult men with micropenises who received hormonal and surgical treatments revealed that persistent dissatisfaction with genital appearance and small penis size in adulthood could considerably impact a person’s sexual quality of life [26]. In patients with 5αRD2, intramuscular T or transdermal DHT gel can be used to improve penile length. As 5αRD2 cannot convert T to DHT, high-dose T therapy has been used to elevate DHT levels and achieve a desirable clinical outcome, and maximum penis enlargement is obtained after 6 months [27]. However, the use of TE in children with micropenises is controversial because of concerns regarding sexual precocity and early epiphyseal closure. This is because T is converted to estradiol, potentially leading to advancement in BA. Consequently, lower doses of TE are recommended in prepubertal patients [4], and low doses of TE for short periods (25 mg every 3–4 weeks for 3–4 times) were thought to not result in the advancement of BA or development of pubic hair [28]. In the present study, sequential TE doses ranging between 25 and 50 mg were administered when the effect was insufficient or in cases of severe micropenises. However, a limited effect was observed, and even low doses administered to prepubertal patients resulted in BA advancement and decreased height SDS. Therefore, caution should be exercised when administering even low doses of T to prepubertal children; furthermore, its usage may be avoided when feasible, especially in patients with 5αRD2, as efficacy cannot be guaranteed. Fortunately, when TE was administered after the age of 10, there was little to no change or an increase in height SDS for BA in three participants. Additionally, both patients who attained their final adult heights after TE treatment received TE treatment after the age of 10; one patient surpassed the MPH in final adult height, and the other patient, although having an adult height below the MPH, showed an increase in height SDS for BA compared with his pre-treatment measurements.

In contrast, DHT transdermal gel offers several advantages over T, including a faster increase in penis size, and is recommended for patients with 5αRD2 [18,29,30,31]. DHT has a much higher potency than T, with approximately four times higher binding affinity for AR and a slower rate of dissociation from AR [32]. The DHT–AR complex is also more readily transformed into the DNA-binding site, and DHT increases AR synthesis while reducing AR turnover [33]. Finally, DHT may be used at higher doses than T, leading to a higher degree of virilization, particularly in patients with 5αRD2 [29]. Sasaki et al. [34] reported that transdermal DHT treatment resulted in a median increment of 2.5 to 0.2 SDS in stretched penile length for patients with prepubertal micropenises.

In addition to its inability to be converted to estrogen by aromatase [2,3], DHT suppresses gonadotrophin and testicular T secretion [2,35], leading to a reduction in the substrate for aromatization to estradiol. The administration of DHT gel reportedly results in a decline in both T and estradiol levels [36], thereby rendering it safe for younger patients in their growth phase [5] as it does not seem to promote bone maturation. However, we lack long-term follow-up studies on the effects of DHT treatment on BA in patients with 5αRD2. Previous studies only tracked patients for a short period of 6 months, with no significant difference in BA observed before and after treatment [18]. In this study, we monitored changes in BA and height SDS for a mean of 5.8 years (from 1.5 to 11.0 years) following DHT treatment. Our results revealed minimal differences in BA compared with that in CA, whereas height SDS for BA increased after DHT treatment. Notably, patients who received DHT treatment before the age of 6–7 exhibited an increase in height SDS for BA, suggesting that DHT treatment during the prepubertal period does not advance BA maturation but supports overall growth, which is beneficial for height.

Notably, DHT treatment has age-limiting effects. For example, post-pubertal penile growth in 5αRD2 patients receiving DHT transdermal treatment was either subtle or arrested [34], and middle-aged volunteers receiving DHT supplementation did not experience marked changes in external genitalia [37]. Therefore, considering the effects of DHT treatment on height and penile length, DHT treatment should be considered during the prepubertal period. Additionally, we suggest administering DHT rather than TE during the prepubertal period. However, TE treatment, with a minimal effect on BA maturation, may be an alternative during the post-pubertal period when DHT transdermal gel is unavailable. Notably, TE administered after the age of 10 did not decrease the height SDS for BA; however, it exhibited a consistent or increasing trend in the current study.

### Limitations

Considering that 5αRD2 is a rare disorder, this study involved a limited number of participants, and their clinical characteristics and treatments received exhibited heterogeneity. As a result, the reliability of our findings may have been compromised and it was difficult to confirm statistical significance. Nevertheless, the extended follow-up period and relatively consistent results may serve as partial compensation for this limitation. Moreover, it is important to note that this study was conducted retrospectively, and certain data, such as the effect of androgen treatment on penis size, were collected subjectively. Although this study did not specifically focus on the effect of androgen treatment on penis size, it was the main purpose and is important for patients with 5αRD2. However, while DHT transdermal cream is widely recognized as the most appropriate treatment option for the pathophysiology of patients with 5αRD2, its effect on height was not adequately evaluated. Further large-longitudinal prospective studies that focus on both penis size and height will be more helpful in considering the appropriate androgen treatment for patients with 5αRD2.

## 5. Conclusions

Collectively, 5αRD2 is a highly rare disease, and androgen treatment is crucial for male gender-assigned patients. We observed that DHT treatment during the prepubertal period did not advance BA and had a positive effect on height. In addition, TE treatment may be considered as an alternative after the age of 10 when DHT treatment is unavailable. Although further studies with larger sample sizes are required to validate our findings, these novel insights would help mitigate the concerns about the adverse effects of androgen treatment on height in patients with 5αRD2.

## Figures and Tables

**Figure 1 jpm-13-00992-f001:**
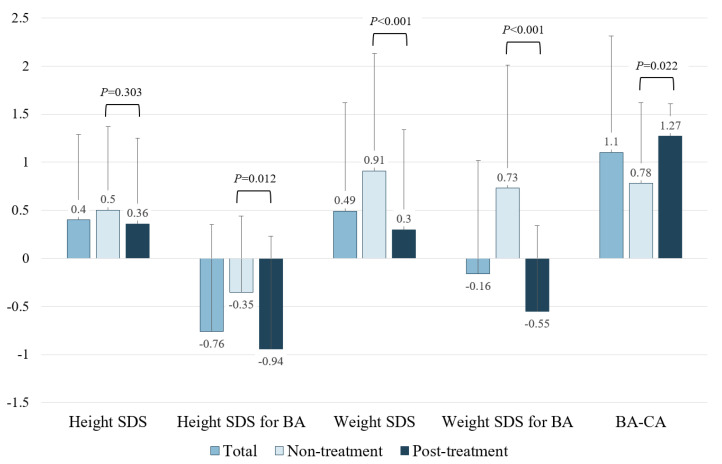
Height and weight standard deviation scores and the comparison between non-treatment and post-androgen treatment groups in 5-α-reductase type 2 deficiency. Each bar represents the mean and standard deviation. Abbreviations: BA, bone age; CA, chronological age; SDS, standard deviation score.

**Figure 2 jpm-13-00992-f002:**
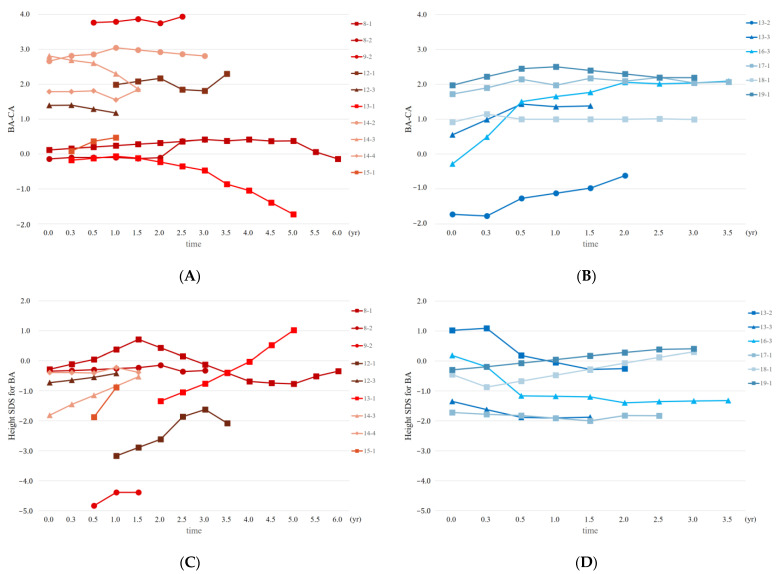
Changes in the difference between BA and CA and height SDS for BA following DHT and TE treatments. (**A**) Changes in the difference between BA and CA following DHT treatment. (**B**) Changes in the difference between BA and CA following TE treatment. (**C**) Changes in the height SDS for BA following DHT treatment. (**D**) Changes in height SDS for BA following TE treatment. The same patients are labeled with the same numbers, and the number of treatments is indicated numerically (such as −1, −2). Abbreviations: BA, bone age; CA, chronological age; SDS, standard deviation score; DHT, dihydrotestosterone; TE, testosterone enanthate.

**Figure 3 jpm-13-00992-f003:**
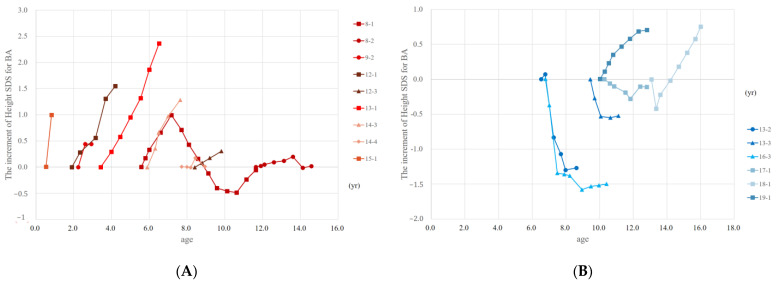
Increment of height SDS for BA based on the start of androgen treatment according to age. (**A**) The increment of height SDS for BA based on the start of DHT treatment. (**B**) The increment of height SDS for BA based on the start of TE treatment. The same patients are labeled with the same numbers, and the number of treatments is indicated numerically (such as −1, −2). Abbreviations: BA, bone age; SDS, standard deviation score; DHT, dihydrotestosterone; TE, testosterone enanthate.

**Figure 4 jpm-13-00992-f004:**
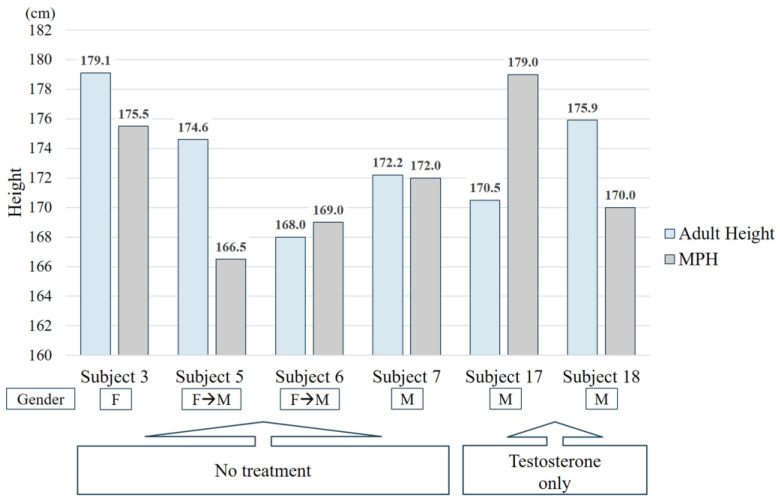
Comparison of final adult height with mid-parental height in 5-α-reductase type 2 deficiency.

**Table 1 jpm-13-00992-t001:** Clinical characteristics of patients with 5-α-reductase type 2 deficiency.

Characteristics (N = 19)		
Current age, yr	11.7 (1.9–27.0)	
Age at the first visit, yr	0.6 (0–16.4)	
Follow-up duration, yr	10.6 (1.4–20.0)	
† Age at molecular diagnosis, yr	7.9 (0–23.7)	
Gender at the first visit, M:F	13:6	
‡ Current gender, M:F	15:4	
EGS at the first visit	4.0 (1.0–11.0)	
Penis		
Micropenis, N (%)	10 (52.6%)	
Clitoromegaly, N (%)	5 (26.3%)	
Clitoris, N (%)	4 (21.1%)	
Hypospadias, N (%)	16 (84.2%)	
Cryptorchidism, N (%)	9 (56.3%)	
Peno-scrotal disposition	5 (26.3%)	
Internal genitalia		
Uterus	0 (0.0%)	
Ovaries	0 (0.0%)	
Hormone levels		
LH (mIU/mL)	1.7 (<0.2–20.7)	
FSH (mIU/mL)	2.0 (0.8–82.3)	
hCG stimulation test		
Baseline T (ng/dL)	120.0 (2.5–623.3)	
Baseline DHT (ng/dL)	9.0 (1.0–48.0)	
Baseline T/DHT ratio	5.2 (0.5–62.1)	
Stimulated T (ng/dL)	378.6 (37.0–1267.0)	
Stimulated DHT (ng/dL)	13.5 (3.0–83.0)	
Stimulated T/DHT ratio	17.4 (7.0–17.4)	
Androgen treatment for penises		
DHT only	5 (26.3%)	Patients nos. 8–12
DHT + Testosterone	4 (21.1%)	Patients nos. 13–16
Testosterone only	3 (15.8%)	Patients nos. 17–19
Male sex assigned and no treatment	1 (5.3%)	Patient no. 7
Gender transition female to male, and no treatment	2 (10.5%)	Patients nos. 5, 6
Female sex assigned and no treatment	4 (21.1%)	Patients nos. 1–4

Abbreviations: N, number; M, male; F, female; EGS, external genitalia score; LH, luteinizing hormone; FSH, follicle-stimulating hormone; hCG, human chorionic gonadotropin; T, testosterone; DHT, dihydrotestosterone. † All patients were confirmed to have mutations in *SRD5A2* and were diagnosed with 5-α-reductase type 2 deficiency. ‡ Two patients were raised as girls; however, after being diagnosed with 5-α-reductase type 2 deficiency, they changed their gender to male. Data are presented as the median (min–max) or N (%).

## Data Availability

Not applicable.

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
