# Peer review of "Effects of Androgen Treatment on Growth in Patients with 5-α-Reductase Type 2 Deficiency"

_jpm, 2023, doi:10.3390/jpm13060992_

Round 1
Reviewer 1 Report
The aim of this study was to describe the effects of DHT or testosterone enanthate (TE) treatment on growth in patients with 5-alpha reductase type 2 deficiency (5αRD2). The subject is interesting because there is really scarce information in the literature. However, there are major methodological issues that need to be addressed:
1. 1. The series of 19 patients is extremely heterogeneous, which precludes from drawing a conclusion:
a. There are patients raised as female and patients raised as male. This raises concern about the way of calculating height SDS: were they all calculated for male (since they were 46,XY), or was it calculated for rearing gender?
b. Patients raised as female did not receive treatment. It may be tricky to compare growth in these patients with those raised as male and receiving treatment. A bias cannot be ruled out, due to a differential severity of the disease.
c. There are patients treated in the first year of life and others at pubertal age. Once again, they are difficult to compare due to the effect of age on height velocity.
d. There are patients who received several cycles and other, only 1 cycle.
In summary, to avoid bias, it would be better if the authors included only patients with similar characteristics (e.g. only those raised as males and of similar age), for comparative analyses.
Alternatively, they could present the study as a case series (including all cases) but refraining from making comparisons when groups are not comparable (e.g. different rearing gender, different ages).
2. 2. If I understood correctly, although patients were followed for years, the authors are assessing a short period of treatment for each patient. This should be made clear to avoid misunderstanding that long treatments (TE and DHT) are being compared.
3. 3. Please provide analysis of variants found in SRD5A2 following the ACMG criteria for pathogenicity. Indicate the meaning of Q6.
4. 4. EMS should be replaced by EGS (van der Straaten, JCEM, 2019).
5. 5. Serum hormone levels should be given for each hormone in addition to hormone ratio (T/DHT).
6. 6. More detailed characterisation of all patients should be given (e.g. FSH, LH, AMH, inhibin B levels; ultrasound or imaging of internal genitalia).
There are minor issues, but they may not remain in the revised version.
Author Response
Please see the attchment

Reviewer 2 Report
The authors describe a long-term follow-up of patients with 5aRD2 treated with either TE or DHT. They monitor the effect on height and bone age and describe different effects depending on the androgen used and on the age at treatment. They come to the conclusion that DHT treatment is preferable over TE treatment especially when used in the prepubertal period. This is an important contribution to the field. Nevertheless, several points have to be addressed prior to publication.
In the abstract the authors state twice comparisons between the non-treated and the androgen-treated group. The corresponding figures/data are missing in the manuscript.
In the introduction the word disease should be replaced by condition in order not to offend patients.
SRD5A2 gene has to be written in italic.
DHT is also important postnatal so it is more correct to delete the word prenatal.
In the Material and Method section it should be stated how the mutations were confirmed. If by Sanger Sequencing, then a chromatogram of the region should be included.
In supplementary table 1, at least the genomic position with the nucleotide change has to be listed (optionally the transcript position with indication of the transcript). Additionally, all missense variants have to be evaluated, if they are pathogenic or not. Most listed variants are known clinvar variants, so it would be sufficient to include the clinvar ID. T53R does not exist. This has to be corrected. If this variant is not listed in clinvar, its pathogenicity has to be evaluated using ACMG criteria.
The sentence: “ The current retrospective study only analyzed the results obtained from the medical records of the clinic; therefore, the requirement for informed consent from participants was waived.” is not clear. It has to be clearly explained why the informed consent was not needed as personal data were collected. A similar sentence at the end of the manuscript has to be explained in more detail as well.
Results:
In table 1 the authors show a summary of the clinical characteristics. Please include the word summary in the first sentence of the results section. In the second sentence there is missing the word median three times.
It is not clear which of the patients 1 – 7 received at some point androgen treatment. This should be specified.
In table 1 legend a N should be specified.
Figure 4B should come before Figure 5.
Figure Legend 6 should be: Comparison of final adult height with mid-parental height (MPH) in patients with 5-α-reductase type 2 deficiency.
Why does Figure 6 not include all patients? A discussion of the results of Figure 6 is missing.
In the conclusions the sentence should read: In addition, TE treatment may be considered as an alternative during the post-pubertal period when DHT treatment is unavailable.
In supplementary table 2 the judgements good, mild etc have to elaborated better. What effect was measured here? EMS? The words good etc should be converted into numbers.
The authors should also discuss to what extend the androgen treatment had an effect on EMS.
Minor editing of English language required
Author Response
Please see the attchment

Round 2
Reviewer 1 Report
Please see the attachment.

Minor editing of English language required
Reviewer 2 Report
The authors reponded largely to the points raised. However the following points have to be addressed before publication:
Suppl figure 1 patient 12 should have a higher resolution. The mutation is not visible in the actual format.
Suppl figure 1 patient 14 is well visible but seem to be copied from another publication. The original data should be presented.
The corresponding transcript (I guess it was NM_000348.4) has to be stated in the legend of suppl. table 2. T53A is still not correct. There is a W at position 53.
see comments to the authors
